# High Levels of Glutaminase II Pathway Enzymes in Normal and Cancerous Prostate Suggest a Role in ‘Glutamine Addiction’

**DOI:** 10.3390/biom10010002

**Published:** 2019-12-18

**Authors:** Thambi Dorai, Bhuvaneswari Dorai, John T. Pinto, Michael Grasso, Arthur J. L. Cooper

**Affiliations:** 1Department of Urology, New York Medical College, Valhalla, NY 10595, USA; thambi_dorai@nymc.edu (T.D.); mgrasso3@earthlink.net (M.G.); 2Department of Biochemistry & Molecular Biology, New York Medical College, Valhalla, NY 10595, USA; john_pinto@nymc.edu; 3Department of Pathology, Montefiore-Nyack Hospital, Nyack, NY 10960, USA; bunadorai@gmail.com

**Keywords:** ω-amidase, glutaminase I, glutaminase II pathway, glutamine transaminase K, α-ketoglutaramate, glutamine addiction

## Abstract

Many tumors readily convert l-glutamine to α-ketoglutarate. This conversion is almost invariably described as involving deamidation of l-glutamine to l-glutamate followed by a transaminase (or dehydrogenase) reaction. However, mammalian tissues possess another pathway for conversion of l-glutamine to α-ketoglutarate, namely the glutaminase II pathway: l-Glutamine is transaminated to α-ketoglutaramate, which is then deamidated to α-ketoglutarate by ω-amidase. Here we show that glutamine transaminase and ω-amidase specific activities are high in normal rat prostate. Immunohistochemical analyses revealed that glutamine transaminase K (GTK) and ω-amidase are present in normal and cancerous human prostate and that expression of these enzymes increases in parallel with aggressiveness of the cancer cells. Our findings suggest that the glutaminase II pathway is important in providing anaplerotic carbon to the tricarboxylic acid (TCA) cycle, closing the methionine salvage pathway, and in the provision of citrate carbon in normal and cancerous prostate. Finally, our data also suggest that selective inhibitors of GTK and/or ω-amidase may be clinically important for treatment of prostate cancer. In conclusion, the demonstration of a prominent glutaminase II pathway in prostate cancer cells and increased expression of the pathway with increasing aggressiveness of tumor cells provides a new perspective on ‘glutamine addiction’ in cancers.

## 1. Introduction

### 1.1. The Glutaminase II Pathway in Mammalian Tissues

Humans, rodents, and other mammals possess two major enzyme/enzyme systems for converting the amide group of glutamine to ammonia. The first—glutaminase I (or phosphate-activated glutaminase (PAG)) simply hydrolyzes glutamine to glutamate and ammonia (Equation (1)). PAG consists of two isozymes, namely a kidney type glutaminase ((GLS1–sometimes written simply as GLS)–and its active shortened form (GAC)) and a liver type glutaminase ((GLS2) and its active shortened form (GAB)). The second—glutaminase II consists of glutamine transaminases with wide α-keto acid specificity (Equation (2)) that transaminate glutamine to α-ketoglutaramate (KGM) coupled to ω-amidase (ω-amidodicarboxylate amidohydrolase), an enzyme that hydrolyzes KGM to α-ketoglutarate and ammonia (Equation (3)). The net glutaminase II pathway is shown in Equation (4). (reviewed in [1]). Humans and rodents possess two major glutamine transaminases: glutamine transaminase K (GTK) and glutamine transaminase L (GTL) [1,2,3,4]. A third transaminase, namely, kynurenine aminotransferase 2 (KAT2) also possesses some activity toward glutamine [1]. Most published work regarding the glutamine transaminases has centered on GTK, as in the current work.
l-Glutamine + H_2_O → l-glutamate + NH_3_(1)
l-Glutamine + α-keto acid ⇆ α-ketoglutaramate (KGM) + l-amino acid(2)
KGM + H_2_O → α-ketoglutarate + NH_3_(3)
Net: l-Glutamine + α-keto acid + H_2_O → α-ketoglutarate + l-amino acid + NH_3_(4)

### 1.2. The Glutaminase II Pathway in Cancer Cells

Only a very few studies of enzymes of the glutaminase II pathway in cancer cells have been published. Meister showed that Novikoff hepatomas exhibit considerable ω-amidase activity [5]. Other workers showed the occurrence of glutamine transaminase activity in a dedifferentiated human astrocytoma [6]. Immunohistochemical studies from our laboratory recently showed intense staining for ω-amidase and GTK in normal human pancreatic, bladder, and prostate cells [1,7]. Human prostate and bladder cancer cells were also shown to exhibit intense staining for GTK and ω-amidase, and human pancreatic cancer cells were shown to intensely stain for GTK [1,7]. Thul et al. [8] have developed a cell atlas using integrated transcriptomic, proteomic, antibody based immunofluorescence microscopy, and mass spectrometry to map the in situ localization of 12,003 proteins at a single cell level. These authors found that ω-amidase (annotated as Nit2 in the human genome) is present in nearly all cell lines examined. Interestingly, the authors demonstrated the presence of ω-amidase, GTK (annotated as kynurenine aminotransferase 1 (KAT1)) and GTL (annotated as kynurenine aminotransferase 3 (KAT3)) in typical cancer cell lines, such as A-431, U-251MG and U2OS [8].

A recent tracer study with N^15^- and C^13^-labeled glutamine showed that the glutaminase II pathway is active in patient-derived pancreatic ductal adenocarcinoma orthotopic tumors implanted in nude mouse pancreas [9]. Evidence was presented that the glutaminase II pathway is upregulated in these cells when GLS1 is inhibited [9]. It was suggested that an inhibitor of the glutaminase II pathway may be clinically effective as an anti-cancer agent [9]. The present work extends our previous work on the glutaminase II pathway in prostate tissue by showing that glutamine transaminase and ω-amidase specific activities are exceptionally high in normal rat prostate and that the glutaminase II pathway may be important for the transfer of α-ketoglutarate as an energy source from supporting stromal cells to cancer cells [1,7].

## 2. Materials and Methods

### 2.1. Chemicals

l-Glutamine, l-methionine, l-selenomethionine (SM), *S*-methyl-l-cysteine (SMC), sodium α-keto-γ-methiolbutyrate (KMB; 4-methylthio-2-oxobutanoate), sodium α-ketoglutarate, β-mercaptoethanol, dithiothreitol (DTT), sodium dodecyl sulfate (SDS) and metaphosphoric acid (MPA) were obtained from Sigma-Aldrich (St. Louis, MO, USA). The α-keto acids derived from SM and SMC (i.e., α-keto-γ-methylselenobutyrate (KMSB) and β-methylselenopyruvate (MSP), respectively) were made in situ from the corresponding l-amino acids with snake venom l-amino acid oxidase [10]. 2,4-Dinitrophenylhydrazine was obtained from Kodak (Rochester, NY, USA). KGM was synthesized from l-glutamine with l-amino acid oxidase as described [11].

### 2.2. Prostate Cancer Tissue Selection

Archived human biopsy specimens in paraffin blocks were obtained from the Pathology Department of New York Medical College (Valhalla, NY, USA). The archival specimens had no identifiable patient information and therefore met the standard for Institutional Review Board exemption. The total number of specimens analyzed for each Gleason grade is 5 (*n* = 5). The recent Gleason grading system including the pathological features is discussed below (Section 3.4).

### 2.3. Prostate Cancer Cell Lines

Four prostate cancer cell lines were used in the present study (i.e., LNCaP, C4, C4-2, and C4-2B). LNCaP was obtained from the American Type Culture Collection (Manassas, VA, USA). All the other cell lines used were obtained originally from ViroMed Laboratories (Minneapolis, MN, USA) and cultured as originally described [12]. The LNCaP cells were originally isolated from a human lymph node metastasis. These cells display androgen signaling and thus exhibit androgen-sensitive growth. When injected subcutaneously into male athymic nude mice, LNCaP cells fail to metastasize to bone. However, Chung and co-workers isolated increasingly metastatic cell lines (C4, C4-2, and C4-2B) from LNCaP cells that were co-cultured with human osteosarcoma fibroblasts [13,14]. Notably, the C4-2B cells have the propensity to develop osseous metastases when injected intra-tibially into severe combined immunodeficiency (SCID) mice and are hormone refractory. Thus, this progression model system recapitulates the development of a localized primary tumor in the prostate of the patient to metastatic invasion of skeletal tissues in the later stages of the disease.

### 2.4. Western Blots

The cultured prostate cancer cells were washed twice in sterile phosphate buffered saline (1× PBS) to remove the medium and were directly solubilized and denatured by adding hot 2× SDS/β-mercaptoethanol-sample buffer (Sigma-Aldrich, St. Louis, MO, USA). The solubilized contents were further denatured by boiling for 5 min. Protein was precipitated with cold acetone to remove interfering substances such as β-mercaptoethanol and estimated by the bicinchoninic acid (BCA) protein assay procedure (as per the instructions in the Pierce BCA Protein Assay Kit, Pierce Biotechnologies, Waltham, MA, USA. Normalized quantities of protein (typically 25 μg) were loaded and electrophoresed on a 12% denaturing polyacrylamide gel. The resolved proteins were blotted onto an Immobilon-polyvinylidene difluoride (PVDF) membrane using standard procedures. The blot was blocked for 1 h with 1× PBS containing 0.1% Tween-20 and 5% nonfat dry milk (BioRad, Hercules, CA, USA). Subsequently, the blot was probed with a rabbit polyclonal antibody to human glutaminase (GLS1; active form, homotetramer; monomer molecular mass, 73.46 kDa) (Novus Biologicals, Centennial, CO, USA), a mouse monoclonal antibody against human Nit2 (i.e., ω-amidase; homodimer; monomer molecular mass ~30.6 kDa) (Origene Biotechnologies, Rockville, MD, USA), a mouse monoclonal antibody against human KAT1 (i.e., GTK; homodimer; monomer molecular mass, ~47.9 kDa) (Santa Cruz Biotechnologies, Dallas, TX, USA) – or a rabbit polyclonal antibody against human housekeeping β-actin (Santa Cruz Biotechnologies, Dallas, TX, USA). Densitometric analyses of the relative levels of expression of GLS1, ω-amidase and GTK in C4, C4-2, and C4-2B cells compared to those in LNCaP cells were performed in triplicate after normalizing for β-actin expression in these immunoblots using the Image Studio Lite software from Li-Cor, Inc. (Lincoln, NE, USA).

### 2.5. Immunohistochemistry

Five-micron thick sections were cut from the paraffin blocks and deparaffinized by consecutive treatments with xylene and graded alcohols. The hydrated sections were washed with PBS for 5 min at room temperature. The slides were immersed in acid citrate (pH 6.0) and the antigen retrieval procedure was conducted at 120 °C for 2 min by means of a pressure cooker, using the protocol provided by Vector Laboratories (Burlingame, CA, USA). Endogenous peroxidase blocking, treatment with normal goat serum, and treatment with primary and secondary antibodies were carried out according to the instructions provided with the Vectastain ABC kit (Vector Laboratories). The primary antibodies used were (1) anti-GLS1; (2) anti-Nit2/ω-amidase, (3) anti-KAT1/GTK (Section 2.4), and (4) a monoclonal antibody against prostatic acid phosphatase (PAP; a specific marker for normal prostate epithelium) from LSBio, Inc. (Seattle, WA, USA). After development of the final color with the diaminobenzidine-based peroxidase staining reagent, the slides were washed, counterstained with hematoxylin QS solution (Vector Laboratories), cleared and mounted with a non-aqueous mounting medium (VectaMount, Vector Laboratories). All primary antibodies were used at a 1:100 dilution using 1 × PBS and dilute normal goat serum. The dilutions of secondary biotinylated antibodies and the wash protocols were exactly as described by the manufacturer of the VectaStain ABC kit (Vector Laboratories). The stained sections were examined under a Nikon Eclipse microscope (E200HD Digital, Melbille, NY, USA) and the images were captured at 20 × magnification. In addition, hematoxylin and eosin (H & E) stains were performed on parallel sections, according to established protocols.

### 2.6. Preparation of Rat Tissues for Enzyme Determinations

Three adult male Sprague-Dawley rats weighing about 350 g were humanely euthanized in a CO_2_ chamber according to the protocols established by the Institutional Animal Care and Use Committee (IACUC) of the New York Medical College. The prostate, liver and kidneys from each animal were isolated and separately homogenized in a five-fold (*w*/*v*) excess of ice-cold 50 mM potassium phosphate buffer (pH 7.4) using a handheld Potter-Elvehjem homogenizer. The crude homogenate was lightly sonicated in a micro sonicator (power setting 3) and then centrifuged at 21,000× *g* to remove cellular debris. The supernatant fraction was then passed through a Centricon filter (30,000 *M*_r_ cutoff, Danvers, MA, USA). The concentrated supernatant fraction was then diluted with 50 mM potassium phosphate buffer (pH 7.4) and concentrated again. This procedure was repeated three more times to ensure removal of endogenous redox compounds (e.g., ascorbate, glutathione, urate) that interfere with electrochemical (EC) detection of redox-active products measured in enzyme-catalyzed reactions (see below).

### 2.7. Measurement of Transaminase Activities with l-Glutamine, MSC, and SM in Rat Tissue Homogenates

Three transaminase activities were measured in the filtered tissue homogenates, namely: (1) l-glutamine–KMB; (2) MSC–KMB; (3) SM–KMB. It was previously shown that the α-keto acid, KMB, is a good co-substrate of both GTK and GTL, whereas MSC and SM are excellent l-amino acid substrates of GTK and GTL, respectively [2,3,4,15]. The reaction mixture (0.25 mL) contained 1 mM l-amino acid, 2 mM KMB, and 25 μL of homogenate in 50 mM potassium phosphate buffer (pH 7.4). The reaction mixture was incubated at 37 °C and at 15-min intervals, 50-μL aliquots were withdrawn and treated with 12.5 μL of 25% *w*/*v* MPA. After centrifugation at 21,000× *g* for 5 min to remove precipitated proteins a 5-μL aliquot of the supernatant fraction was analyzed by high performance liquid chromatography (HPLC) with EC detection as described previously [10,16,17]. Under these conditions the retention times of l-methionine, MSC, SM, KMSB, and MSP are 3.7, 6.0, 6.8, 4.1, and 4.7 min, respectively. For each tissue homogenate analyzed, the reaction was found to be linear over a 40-min period. Thus, 40 min of incubation at 37 °C was chosen for determination of each transaminase activity. Both KMB and l-methionine are redox positive. Thus, transamination between l-glutamine and KMB can theoretically be determined by measuring appearance of l-methionine or disappearance of KMB. In independent experiments it was shown that the two measurements give comparable results. In the case of reaction mixtures containing MSC/KMB or SM/KMB, both l-amino acid substrates are also redox active as are the α-keto acid products. Thus, for these substrate pairs, enzyme activity can theoretically be determined by disappearance of l-amino acid substrate, disappearance of KMB, appearance of l-methionine or appearance of α-keto acid product. In separate experiments it was shown that the concentration of all products formed matched the concentration of substrate consumed for the MSC/KMB and SM/KMB reaction mixtures. In practice, however, in the current work, since KMB was the common substrate in all transamination reactions, l-methionine appearance was used to maintain continuity in product formation and to compare the reaction rates in tissues when glutamine, MSC or SM were used as co-substrates.

### 2.8. Measurement of ω-Amidase Activity in Rat Tissue Homogenates

ω-Amidase activity was measured directly in an aliquot of the crude rat tissue homogenate. After freeze-thawing the homogenate twice, the homogenate was diluted ten-fold with 50 mM potassium phosphate buffer (pH 7.4) and then a 2-μL aliquot was assayed for enzyme activity by the method of Krasnikov et al. [11]. The reaction mixture contained, 5 mM KGM, 100 mM Tris-HCl buffer (pH 8.5), and 2-μL tissue homogenate in a final volume of 50 μL. After incubation for 30 min at 37 °C α-ketoglutarate was measured as its 2,4-dinitrophenylhydazone in an end-point assay [11]. In preliminary experiments it was shown that α-ketoglutarate production catalyzed by the prostate preparation is linear for at least 30 min.

### 2.9. Statistics

Data are reported as the mean ± SD. Differences between transaminase and ω-amidase specific activities in rat prostate versus those in rat liver and kidney were determined by the two-tailed unpaired *t*-test using on-line GraphPad software. A *p* value of ≤0.05 is considered significant.

## 3. Results

### 3.1. Transamination of l-Glutamine, SM, and MSC Catalyzed by Preparations of Rat Liver, Kidney, and Prostate

Results of the transamination assays are summarized in Table 1. Previously it was noted that, of all ten rat tissues investigated, the highest glutamine transaminase specific activity was with liver and kidney [3]. Prostate was not investigated in that study. Moreover, in that study, transamination was measured between 20 mM l-glutamine and 0.4 mM phenylpyruvate, whereas in the present study transamination was measured between 1 mM l-glutamine and 2 mM KMB. Nevertheless, the data indicate that, under the present assay conditions, the rat prostate possesses significantly higher glutamine transaminase specific activities than does the rat kidney and liver. As noted above, MSC and SM are excellent substrates of GTK and GTL, respectively [4]. Interestingly, the specific activities obtained with these substrates are again significantly much higher in rat prostate than in rat liver or kidney.

Three transaminase specific activities (T) were determined in homogenates that had been subjected to ultrafiltration to remove small *M*_r_ molecules and proteins with *M*_r_ values < 30,000. The specific activities of the three transaminase reactions in rat prostate are higher than those in rat liver and kidney with *p* ≤ 0.001 for all cases. As noted in the text, MSC and SM are good substrates of GTK and GTL, respectively. Indeed, western blotting (Figure 1) confirms the presence of GTK in human prostate cancer. The possible presence of GTL in human prostate cells is yet to be directly determined. The specific activity of ω-amidase was determined directly on 2-μL aliquots of 10-fold diluted crude homogenates; *p* = 0.06 for difference in the specific activity of ω-amidase in rat prostate versus that of rat liver, and *p* = 0.02 for the difference in specific activity of ω-amidase in rat prostate versus that of rat kidney. Tissue specimens were obtained from three individual rats. Enzyme assays were carried out in triplicate for each tissue specimen. The results are expressed as mean ± standard deviation.

### 3.2. ω-Amidase Activities in Homogenates of Rat Liver, Kidney and Prostate

The findings are summarized in Table 1 as shown above. The difference in ω-amidase specific activities between rat prostate and rat liver did not quite reach significance (*p* = 0.06). However, the specific activity of ω-amidase is slightly, but significantly (*p* = 0.02), lower in rat prostate than in rat kidney. Nevertheless, the specific activity of ω-amidase is remarkably high in rat prostate. Thus, the finding of relatively high specific activities of both glutamine transaminase and ω-amidase in rat prostate suggests that this tissue possesses a highly active glutaminase II pathway, comparable to that of liver and kidney. These tissues were previously thought to have the highest glutaminase II specific activity in rat tissues [2,3].

### 3.3. Western Blotting of Cell Lysates of Various Prostate Cancer Cells

Western blot analysis showed that, in the progression of human prostate cancer cells from least aggressive (LNCaP cells) to most aggressive (C4-2B cells), a progressive increase occurs in the intensity of staining for GLS1 relative to that of β-actin (Figure 1). Interestingly, a marked increase can also be observed in the intensity of ω-amidase staining and GTK staining (Figure 1).

These results indicate that there is a progressive and significant increase in the levels of these enzymes over those expressed in the androgen-responsive LNCaP cells. Densitometric analyses of the relative levels of expression of these enzymes in these immunoblots are shown Figure 2.

These results indicate that a progressive and significant increase occurs in the levels of the enzymes of the glutaminase II pathway over those that are expressed in the androgen-responsive LNCaP cells. These results also suggest that in the malignant prostate, there is a remarkable metabolic heterogeneity in terms of glutamine utilization, and that both GLS1 and the ω-amidase/GTK couple may play a critical role in prostate cancer progression and/or in the establishment of osseous metastases.

### 3.4. Immunohistochemical Staining of GLS1, ω-Amidase, and GTK in Normal Human Prostate and in Cancerous Human Prostate of Increasing Aggressiveness

Studies examining the glutaminase II pathway in prostate cell lines were extended to include archived and de-identified prostate cancer specimens. Normal prostate tissue and cancerous prostate tissue with varying degrees of cancer progression (Gleason grades) were investigated to determine whether our findings with isolated rat prostate (Table 1) and immunoblotting of human prostate cancer cell lines (Figure 1 and Figure 2) are reflected in in vivo conditions. The results are shown in Figure 3a, which depicts the H & E stain of the normal human prostate gland and that of prostate cancers arranged according to the newly revised Gleason pattern scoring system of grades 1–5, with grade 5 being the most advanced cancer.

These images illustrate the typical and well-characterized pathological features consistent with Gleason grading. For example, Gleason grades 1–3 specimens show closely packed and variably sized glands. Gleason grade 4 glands show large irregular masses of neoplastic cells with cribriform glandular cells, which display some fusion. The advanced Gleason grade 5 pattern shows sheets of fused cancer cells which are poorly differentiated, and anaplastic, with a complete lack of glandular architecture. The immunohistochemical staining for GLS1, ω-amidase and GTK carried out on parallel sections reveals an interesting pattern. The normal glandular cells of the prostate stain intensely for GLS1, ω-amidase and GTK. In the normal prostate, the GLS1 enzyme is localized only in the glandular structure and no GLS1 stain is detected in the stromal cell compartment (Figure 3b). However, as the Gleason grade increases from 3 to 5 the GLS1 staining in the glandular compartment is maintained, but GLS1 staining in the stromal cell compartment now becomes apparent, and increases with increasing Gleason grade. The staining intensity of ω-amidase and the GTK enzymes follows a similar pattern (Figure 3c,d, respectively). The diffuse stain seen in the stromal cell compartment may be due to an invasion of the epithelial cancer cells into this compartment or to an intrinsic upregulation of expression of these enzymes by oncogenic signaling in the stromal cell compartment. To distinguish between these two possibilities, we performed a set of immunohistochemical staining experiments with the monoclonal antibody for PAP, which is a specific marker for glandular epithelium (Figure 3e). In this panel, the PAP stain is localized to the glandular epithelial cells with negligible signal in the stromal cell compartment. This observation strongly suggests that there is negligible invasion of the cancer epithelial cells into the stromal cell compartment and hence the increased signals for GLS1 and ω-amidase must be due to other mechanisms (see Section 4). However, it is important to note here that the glutaminase I pathway (GLS1) and the alternative and/or compensating glutaminase II pathway (glutamine transaminase plus ω-amidase as a functional couple) are very highly represented in normal as well as in malignant human prostate.

## 4. Discussion

### 4.1. The Glutaminase II Pathway Provides Anaplerotic a-Ketoglutarate in Normal and Cancerous Prostate while Closing the Methionine Salvage Pathway

Many cancer cells are ‘addicted’ to glutamine as an anaplerotic source of α-ketoglutarate while at the same time providing nitrogen for the synthesis of key small *M*_r_ metabolites and DNA (e.g., [9,18,19,20]). As noted in the abstract, the conversion of l-glutamine to α-ketoglutarate in cancer cells is almost invariably ascribed to the glutaminase I pathway: l-glutamine is hydrolyzed to l-glutamate followed by conversion of l-glutamate to α-ketoglutarate by an aminotransferase (transaminase) or by glutamate dehydrogenase. However, as also noted, mammalian tissues possess another pathway by which l-glutamine is converted to α-ketoglutarate, namely the glutaminase II pathway, consisting of a glutamine transaminase coupled to ω-amidase (Equations (1)–(4)). In this context, tracer studies with [^15^N]glutamate, [2-^15^N]glutamine, [5-^15^N]glutamine and [2,5-^15^N]glutamine strongly suggest that the glutaminase II pathway plays a prominent role in the metabolism of l-glutamine in humans [21]. Despite this finding, the glutaminase II pathway has been largely overlooked by biochemists and cancer biologists, even though this pathway is ubiquitous in nature [1].

The glutaminase II pathway is also of interest because of its linkage to polyamine synthesis and the methionine salvage pathway. Prostate cells synthesize and secrete large amounts of polyamines and acetylated polyamines into the seminal fluid ([22,23,24] and references cited therein). Carbons 2–4 of methionine and the amine nitrogen are used in the synthesis of putrescine, spermidine and spermine–carbon 1 is lost as CO_2_. A product of the polyamine synthesis pathway is 5′-methylthioadenosine (MTA). Loss of methionine during polyamine synthesis is counterbalanced by the synthesis of methionine from MTA via the methionine salvage pathway. In this pathway, the original sulfur and methyl of methionine are retained, whereas carbons 1–5 are formed anew from the ribose portion of MTA [25,26]. However, what is the origin of the amine in the salvaged methionine? Early work showed that the last step of the methionine salvage pathway in liver homogenates involves transamination of KMB to methionine and that glutamine is an effective amine donor in this process [27]. Indeed, KMB is a good α-keto acid substrate of purified GTK/KAT1 and GTL/KAT3 enzymes [1,2,3,15,28,29,30]. This finding emphasizes the fact that transamination of KMB with glutamine (coupled to the ω-amidase reaction) is useful to the cell for at least two reasons. ***Not only does this process close the methionine salvage pathway, but it simultaneously provides anaplerotic α-ketoglutarate to the cell.*** This is exemplified by Equation (5) which is a recast of Equation (4), in which the general terms, α-keto acid and l-amino acid, are replaced by KMB and l-methionine, respectively. This process is hypothesized to be important in many cancers, but especially in prostate cancer cells which possess a very active methionine salvage pathway (e.g., [31] and references cited therein). In this context, it is important to note that the methionine salvage pathway is ubiquitous in nature and is known to have arisen early in evolution [26]. Moreover, the glutaminase II pathway (i.e., glutamine transaminase plus ω-amidase) is now well established in bacteria and plants to potentially act in tandem with the methionine salvage pathway [32]—a linkage that apparently has been preserved in mammals during evolution.
l-Glutamine + α-keto γ-methiolbutyrate (KMB) + H_2_O [***closure of the methionine*** [***salvage pathway***] → α-ketoglutarate [***anaplerosis***] + l-methionine + NH_3_(5)

### 4.2. Role of the Glutaminase II Pathway in Replenishing Citrate Carbon in Normal and Cancerous Prostate

Human prostate has an enormous capacity to synthesize citrate. For example, Kline et al. [33] reported citrate concentrations in semen and expressed prostatic secretions from cancer-free human males of 132 ± 30 and 222 ± 55 mM, respectively, but somewhat lower concentrations in patients with prostate cancer (48 ± 8 and 82 ± 36 mM, respectively). Prostate epithelial cells express high levels of the zinc transporter Zip1, allowing the prostate to accumulate zinc – an aconitase inhibitor ([33,34,35] and references cited therein). As a result, citrate accumulates in these cells and is excreted in the seminal fluid, but a significant portion also acts as a source of acetyl-CoA, which can be used by the prostate to generate lipids, cholesterol, and steroid hormones [36]. Zip1 is downregulated in prostate cancer cells and thus the mitochondrial aconitase is disinhibited [36]. Thus, the seminal fluid from prostate cancer patients contains less citrate than that of normal human males [36]. The downregulation of Zip1 permits in the cancer cells relative to normal prostate cells (1) more effective utilization of citrate as an energy source, and (2) more effective lipid production [36].

As pointed out by Eidelman et al., the shortfall in carbon flux through the TCA cycle in the normal prostate as a result of citrate excretion is compensated in part by avid uptake of glutamine as an anaplerotic source [35]. In this context, GLS1 is present in normal and cancerous human prostate, and its expression increases with increasing aggressiveness of the prostate cancer cells [37]. Our data are in accord with this finding (Figure 1). Increased GLS1 activity when coupled to the glutamate dehydrogenase/glutamate-linked aminotransferase reactions would provide anaplerotic carbon to replenish the TCA cycle. However, it should be noted that our findings (Table 1; Figure 1, Figure 2 and Figure 3) also show that the glutaminase II pathway (i.e., Gln → KGM → α-ketoglutarate) is exceptionally well represented in rat prostate and cancerous human prostate, suggesting an important role for the glutaminase II pathway in providing anaplerotic α-ketoglutarate to these cells.

### 4.3. The Glutaminase II Pathway Permits the Formation of a-Ketoglutarate from l-Glutamine under Hypoxic/Anoxic Conditions

GTK is highly expressed in epithelial tissues. For example, in the brain, GTK (and ω-amidase) is present in relatively high levels in the choroid plexus [38]. In the kidney, GTK is strongly expressed in the proximal tubular epithelia [39]. GTK and ω-amidase are also present in normal and cancerous human bladder epithelial cells and in normal and cancerous human prostate epithelial cells [1,7] (present work). We suggest that the relatively high concentrations of GTK and ω-amidase in epithelial tissues, the high concentration of glutamine in the circulation (~0.5–0.8 mM [40]) and the presence of high capacity glutamine transporters (e.g., [41,42]) are important for maintaining cellular nitrogen and energy homeostasis. The glutaminase II pathway may be especially important in rapidly dividing cells that are known to consume glutamine as an important energy source, such as enterocytes [43], lymphocytes [44], and bone cells [45]. Our findings [1,7] (present work) together with the findings of Udupa et al. [9] suggest that the glutaminase II pathway is also important in rapidly dividing cancer cells. Interestingly, the glutamine transporter ASCT2 is strongly expressed in highly proliferative cells, including cancer cells [41,42].

The glutaminase II pathway requires the presence of a suitable supply of α-keto acid substrate for the glutamine transaminases. As noted in Section 4.1, one such α-keto acid is KMB, produced ***endogenously*** during operation of the methionine salvage pathway. It is also highly probable that α-keto acid substrates of the glutamine transaminase reaction are obtained exogenously from the circulation. In this context, α-keto acid substrates such as pyruvate and the branched-chain α-keto acids are present in human serum in the range of 10–100 μM [46]. Moreover, it has been shown that KMB and the branched-chain α-keto acids are transported across the blood-brain barrier, possibly on the monocarboxylate carrier [47]. Interestingly, simultaneous perfusion of isolated rat liver with l-glutamine and α-keto acids (KMB, phenylpyruvate, or the branched-chain α-keto acids) resulted in rapid formation of the corresponding l-amino acids, attesting to the high capacity of the glutaminase II pathway in liver [48]. A recent study showed that α-ketoisocaproate is taken up by glioblastoma cells [49]. This may be advantageous to the cancer cell. Inspection of Equation (4) shows that production of α-ketoglutarate from glutamine via the glutaminase II pathway does not involve a net oxidation. ***Thus, provision of α-keto acid substrates of glutamine transaminase(s) ensures that conversion of glutamine to α-ketoglutarate via the glutaminase II pathway will operate effectively in hypoxic/anoxic regions of the tumor***.

### 4.4. Biological Importance of GLS1 and Glutaminase II Pathway Enzymes in the Stromal Cell Compartment of Human Prostate Cancer Cells

The present work shows the presence of enzymes of both the GLS1 and the glutaminase II pathways in stromal cells in human prostate cancer, but not in normal human prostate. Although we cannot completely exclude the possibility of direct upregulation of gene expression within these cells, our data strongly suggest that the elevated enzyme activities may be due to exocytosis of large extracellular vesicles (LEVs) derived from epithelial cancer cells that contain these enzymes as cargo. Thus, as shown in Figure 3e, the PAP stain is confined to the sheets of epithelial cells, whereas the stromal cell compartment although clearly visible by H & E staining is devoid of the PAP stain. Moreover, as demonstrated in an independent study from our laboratory, metastatic human prostate cancer cells actively discharge LEVs into the tumor microenvironment [50]. Furthermore, GLS1 is a constituent of these LEVs [50]. Thus, LEVs have the opportunity to enter the tumor microenvironment and fuse with stromal cells, thereby metabolically reprogramming (or ‘rewiring’) this compartment (cf. [51,52,53]). Considerable evidence suggests that this metabolic rewiring of metabolism in stromal cells contributes to cancer malignancies [51,52,53,54,55,56,57,58,59]. Our results suggest that as the Gleason grade increases, rewiring of glutamine metabolism in the stromal cell compartment contributes to advancing malignancy.

### 4.5. On the Relative Affinities Exhibited by GLS1 and GTK toward l-Glutamine and Relevance to the In Vivo Metabolism of l-Glutamine

Human KAT1/GTK and mouse KAT3/GTL exhibit apparent *K*_m_ values of 2.8 mM [29] and 0.7 mM [30], respectively, toward glutamine [29]. Inasmuch as glutamine is the most abundant amino acid in the body and the concentration of this amino acid in most tissues is in the mM range (for example the concentration of glutamine in a normal 70 kg man has been estimated to be ~7 mmol/kg (~8 mM) [40,60]), the inherent in vivo capacity of the glutamine transaminases must be considerable. Botman et al. reported a *K*_m_ value for glutamine in mouse kidney and brain (i.e., for the GLS1 isozyme) of 0.6 mM and that for glutamine in mouse liver (i.e., for the GLS2 isozyme) of 11.6 mM [61].

The above discussion does not take into account relative *V*_max_ values. However, the findings suggest that glutamine transaminase activity may be at least comparable to that exhibited by the two glutaminase isozymes. This conclusion is reinforced by the findings of Darmaun et al. [21]. As noted above (Section 4.1), tracer studies conducted by these authors strongly suggest a prominent role for the glutaminase II pathway in humans. Indeed, ***the whole body turnover of glutamine via glutamine transamination may have been even greater than that exhibited by the glutaminase (PAG) reaction*** [21]. These findings have been largely overlooked in the literature, but are consistent with a prominent role for the glutaminase II pathway for the metabolism of glutamine in humans. Moreover, both GTK and GTL are present in the cytosol and mitochondria ([1] and references cited therein; [28]) whereas GLS1 and GLS2 are predominantly mitochondrial. Thus, the glutaminase II pathway has a compartmental advantage over that of the glutaminase I pathway in that it occurs in both the cytosol and mitochondria, whereas the glutaminase I pathway is confined largely to mitochondria.

### 4.6. Glutaminase II Pathway Inhibitors May Offer Potentially Significant Therapeutic Opportunities

Several clinical trials are currently under way to test the efficacy of GLS1 inhibitors as anti-cancer agents [62]. The present work and that of Udupa et al. [9] suggest that inhibitors of the glutaminase II pathway may be effective anti-cancer agents, perhaps in combination with those available for GLS1. In this regard, Ogier et al. [63] and Quash et al. [64] have developed transition state (TS) mimetics that inhibit KMB transamination in cells in culture. These TS mimetics include l-methionine ethyl ester pyridoxal (MEEP). Inasmuch as KMB is a good substrate of both GTL and GTK, it is probable that these TS mimetic inhibitors can block the glutamine transaminases. In the Quash et al. study, MEEP was shown to induce DNA strand breaks in HeLa cells typical of apoptotic cell death [64].

Because kynurenate is a neuromodulator, the pharmaceutical industry is interested in the development of KAT inhibitors [65]. Some inhibitors have been designed to inhibit GTK/KAT1 (based on kynurenine as amino acid substrate) with *K*_i_ values ranging from ~20 μM to ~1 mM [63], but to our knowledge these have not been evaluated as anti-cancer agents. The pharmaceutical industry has yet to focus on designing effective inhibitors of GTL/KAT3. Another strategy for inhibiting the glutaminase II pathway may be to develop selective ω-amidase inhibitors. However, to our knowledge, no potent and selective inhibitors of this enzyme have been described.

## 5. Conclusions and Future Directions

In conclusion, the present work emphasizes the important role of the glutaminase II pathway (i.e., glutamine transaminases coupled to ω-amidase) in prostate biology. Evidence is presented that the glutaminase II pathway in normal and cancerous prostate cells is linked to polyamine and methionine salvage pathways, and that it is an important source of anaplerotic α-ketoglutarate. The present work also suggests that the glutaminase II pathway enhances the metabolic rewiring in the stromal cells, thereby contributing to the malignancy associated with advanced prostate cancer. Finally, it will be important to develop potent and selective inhibitors of GTK, GTL, and ω-amidase as possible anti-cancer agents, possibly to be administered adjunctively with GLS1 inhibitors already under clinical evaluation.

## Figures and Tables

**Figure 1 biomolecules-10-00002-f001:**
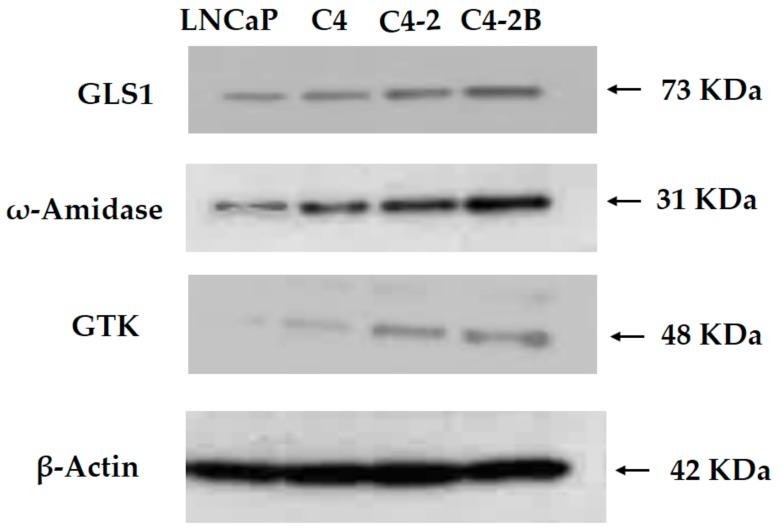
Western blot of enzymes involved in the metabolism of glutamine in human prostate cancer cells. The housekeeping protein β-actin was probed in parallel to ensure equivalent loading of samples for probing of other protein markers such as GLS1, ω-amidase, and GTK. Progression of aggressiveness of the cancer cells is in the order: LNCaP < C-4 < C4-2 < C4-2B.

**Figure 2 biomolecules-10-00002-f002:**
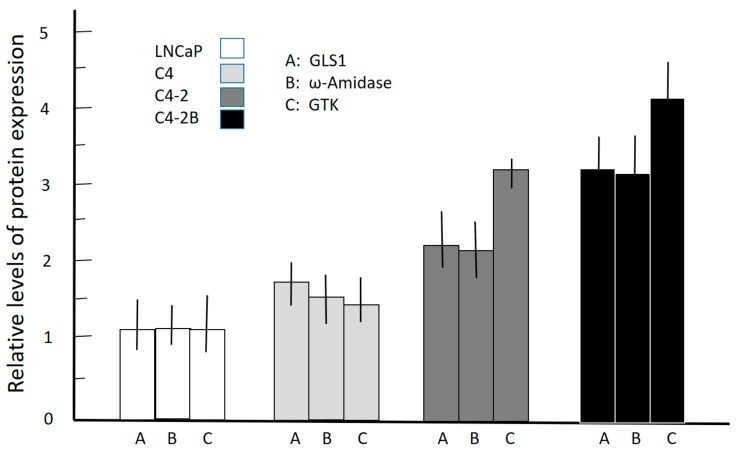
Densitometric analysis of the relative levels of expression of GLS1, ω-amidase, and GTK in C4, C4-2, and C4-2B cells compared to their expression in parental LNCaP cells after normalizing for β-actin expression. The results are expressed as mean ± standard deviation obtained from three independent experiments.

**Figure 3 biomolecules-10-00002-f003:**
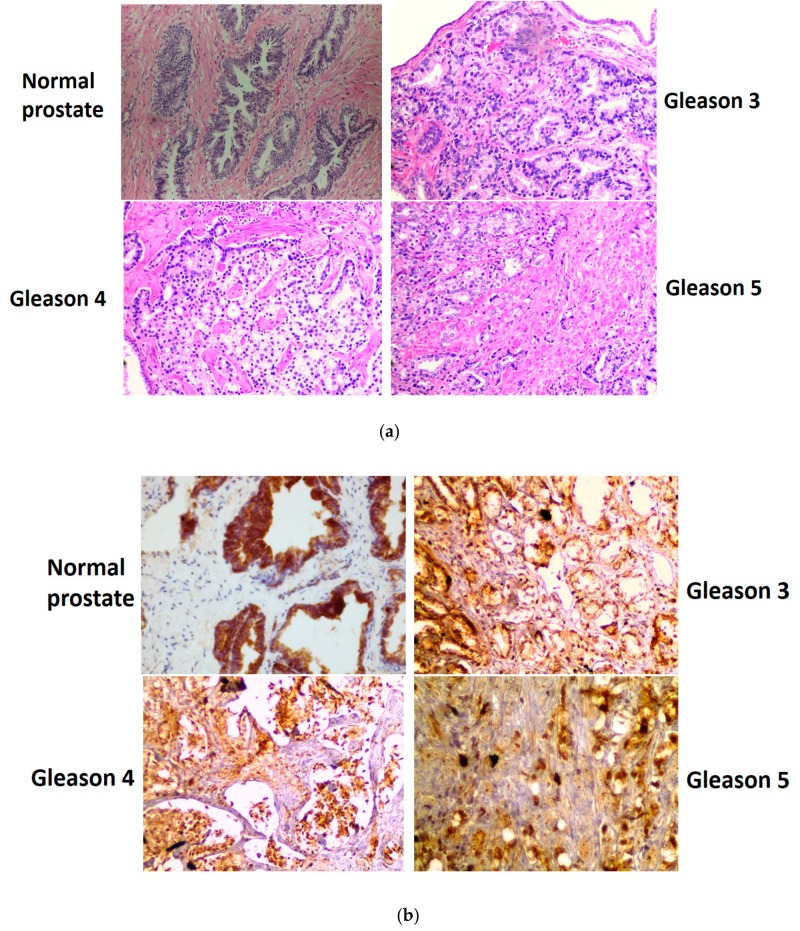
Representative H & E staining (**a**) and immunohistochemical staining for glutaminase GLS1 (**b**), ω-amidase (**c**), GTK (**d**), and for prostatic acid phosphatase (PAP) (**e**) in sections of normal human prostate and prostate cancers of increasing Gleason grade (i.e., increasing aggressiveness). Note the high levels of expression of GLS1, ω-amidase, and GTK in the glandular architecture of the normal human prostate. In normal prostate, these enzymes have negligible expression in the stromal cell compartment. However, in cancerous human prostate tissues, these enzymes are expressed in the stromal cell compartment and the staining intensity increases concomitantly with increasing Gleason grade. To test the cellular origin of the diffuse positive signal in the stromal cell compartment, an antibody was used against the prostate epithelial cell marker (i.e., PAP) on a Gleason 3 prostate cancer pathological specimen (Figure 3e, left panel), which shows the restriction of the signal to the glandular epithelial cells with negligible signal in the stromal cell compartment. The panel on the right shows the corresponding negative control without the antibody where the epithelial and the stromal cell compartments are clearly discernible. See the text for the interpretation of these pathological findings.

**Table 1 biomolecules-10-00002-t001:** Transaminase (T) and ω-amidase specific activities (nmol/min/mg protein) in rat tissue homogenates

	Gln-KMB (T)	MSC-KMB (T)	SM-KMB (T)	ω-Amidase
Liver	0.28 ± 0.08	0.23 ± 0.02	0.48 ± 0.08	97 ± 11
Kidney	0.30 ± 0.02	0.31 ± 0.07	0.43 ± 0.02	102 ± 20
Prostate	0.81 ± 0.08	0.81 ± 0.07	0.90 ± 0.05	66 ± 2

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
