# Peer review of "High Levels of Glutaminase II Pathway Enzymes in Normal and Cancerous Prostate Suggest a Role in ‘Glutamine Addiction’"

_biomolecules, 2019, doi:10.3390/biom10010002_

Round 1
Reviewer 1 Report
The manuscript "High Levels of Glutaminase II Pathway Enzymes in Normal and Cancerous Prostate and their Proposed Role in “Glutamine Addiction” has been revised accordingly with the request. The Authors have addressed the most critical points impruving significantly the scientific soundness of the work. In general, I appreciated the revision performed by the Authors.
Reviewer 2 Report
Unfortunately, I cannot not follow the changes made by the authors. As you will see below, the authors declare that they have expanded Figures 4 a, b, and that they corrected a typo.
Neiher the figures nor the paragraph on page 12 (orignal version) can be found in the revised version.
-Figs. 4a and 4b: The cycles including the structural formulas and their labeling are too small even though the authors’ reasonable attention might be to draw the reader’s attention to the differences (between a=normal and b=malignant).
Answer: We have also noted the same problem with the depiction of the metabolic cycles that include several structural formulae. Because of the condensed nature of this Figure, they tend to become too small and hence difficult to read.
To correct this, we have expanded the Figures to a larger scale so that the readers can now view and appreciate the biochemical alterations that go on in normal versus cancerous prostate.
-Page 12, line 415 "..almost all such studies of cancer addiction in cancer cells assume…". Probably, there is a typo in that the first "cancer" should probably be "glutamine".
Answer: In Page 12 and in line 415, the typo has been corrected.
This manuscript is a resubmission of an earlier submission. The following is a list of the peer review reports and author responses from that submission.
Round 1
Reviewer 1 Report
The ms. by Thorai et al. concerns the expression of Glutaminase II pathway in normal and neoplastic prostate. The contributon is simple but well structured and provides interesting data on a relatively unexplored aspects of prostate metabolism. I have some remarks:
1) Figs. 1 and 2. Apparently the expression of enzymes involved in both Glutaminase I and II pathway is increased in parallel during the progression in vitro of prostate cancer cells. Since these pathways are in competition, what can be the biological significance of a parallel activation ?
2) Figure 3. How can the authors be sure that the diffuse positivity (of both pathways) they note in cancerous prostate is effectively due to a stromal expression and not to invasive malignant epithelial cells? Is it possible to check the co-expression of epithelial or mesenchymal markers with the enzymes?
3) What is the apparent Km of glutaminase II enzymes for glutamine and asparagine? As far as glutamine is concerned, is it lower or higher than that of GLS1?
4) In the discussion, authors discuss the translational potential of glutaminase II pathway inhibitors? Do these inhibitors exist? Are they specific for these enzymes or also interact with other metabolic pathways involved in Gln metabolism?
Author Response
Reviewer 1
The ms. by Thorai et al. concerns the expression of Glutaminase II pathway in normal and neoplastic prostate. The contributon is simple but well structured and provides interesting data on a relatively unexplored aspects of prostate metabolism. I have some remarks:
Answer: We thank the reviewer for the encouraging statements and for his/her constructive criticisms.
1) Figs. 1 and 2. Apparently the expression of enzymes involved in both Glutaminase I and II pathway is increased in parallel during the progression in vitro of prostate cancer cells. Since these pathways are in competition, what can be the biological significance of a parallel activation ?
Answer: We suggest that since most tumors exhibit avid glutamine addiction it makes sense for the “addicted” tumor to maximize the effectiveness of converting glutamine to α-ketoglutarate by activating both pathways. In addition, we believe that there are added advantage to using the glutaminase II pathway. The glutaminase I pathway may be depicted as: Gln " Glu " α-ketoglutarate. The last step may be carried out by glutamate-linked aminotransferases (transaminases) or by the glutamate dehydrogenase (GDH) reaction. However, the GDH-catalyzed conversion of glutamate to α-ketoglutarate is an oxidative event, which would be highly restricted in the hypoxic environment in the tumor. On the other hand the glutaminase II pathway, which may be depicted as Gln " α-ketoglutaramate " α-ketoglutarate, can occur under extremely hypoxic conditions. Moreover, because the prostate synthesizes a large amount of polyamines, the methionine salvage pathway is of high activity in the prostate (Bistulfi et al. Oncotarget 7, 14380-93, 2016). Closure of the methionine salvage pathway involves transamination of α-keto-γ-methiolbutyrate (KMB). Glutamine is an important partner for transamination of KMB. Thus, the glutaminase II pathway “kills two birds with one stone”. It provides anaplerotic α-ketoglutarate to the TCA cycle and at the same time contributes to the closure of the methionine salvage pathway.
2) Figure 3. How can the authors be sure that the diffuse positivity (of both pathways) they note in cancerous prostate is effectively due to a stromal expression and not to invasive malignant epithelial cells? Is it possible to check the co-expression of epithelial or mesenchymal markers with the enzymes?
Answer: We have observed the “diffuse positivity” of both pathways (as observed by immunohistochemistry) in the stromal cell compartment on a number of human prostate cancer specimens. This diffuse nature of the enzyme staining could arise theoretically by at least 3 different mechanisms: 1) As the reviewer suggests, it could be due to prostate cancer epithelial cells that invade the stromal cell compartment. 2) It could also be due to an inherent activation of gene expression of both these pathway enzymes in the stromal cell compartment. 3) Finally, it could be that these epithelial cancer cells shed large extracellular vesicles (LEVs) containing GLS1, ω-amidase and GTK as part of their cargo, which then fuse with the stromal cells and give rise to such a diffuse positivity. To distinguish among these possibilities, we have recently performed another set of immunohistochemistry studies on the human pathological samples that are on hand. We have included these “new” staining pictures along with Figure 3. In this set of extra pictures, we show the staining pattern for the marker PAP (prostatic acid phosphatase) which is a well-established marker for glandular epithelium. Please note in this figure that the PAP stain is still confined to the sheets of epithelial cells while the stromal cell compartment is clearly visible and devoid of the PAP stain. This observation argues against the possibility of epithelial cell invasion as the reviewer suggested. In the negative control, staining without the antibody, the epithelial and the stromal cell compartments are very clearly discernible. Moreover, in an independent study published last year, we clearly showed that metastatic prostate cancer cells (as exemplified by the C4-2B prostate cancer progression model system) do actively shed LEVs into the tumor microenvironment (Dorai et al, Prostate 78, 1181-95, 2018). Interestingly, in this particular study, we showed that both GLS1 and transglutaminase (TG2) are shed into these LEVs. Thus, these LEVs have the opportunity to enter the tumor microenvironment (TME) and fuse with the cells in that compartment, namely the stromal cells, with the possibility of activating that compartment by metabolic reprogramming. This mode of activation of the stromal cell compartment is well established by now by several investigators. Please see relevant references in Minciacchi et al. Oncotarget 6, 11327-41, 2015 and Wendler et al. Trends Cancer 2, 3269, 2016. Thus, while we believe that the diffuse positivity that we observe in our studies could be due to the LEVs that are shed, carrying GLS1 and other enzymes as cargo and fusing these LEVs with the stromal cell compartment, we cannot exclude the last possibility that the diffuse positivity could arise from an inherent upregulation of gene expression of these pathway enzymes by enhanced oncogenic signaling mechanisms that occur in malignant prostate. We admit that we have not yet studied the possible LEV-mediated extrusion of the glutaminase II pathway enzymes (ω-amidase and GTK) in advanced prostate cancer. We will certainly do so in the future to address these interesting possibilities.
3) What is the apparent Km of glutaminase II enzymes for glutamine and asparagine? As far as glutamine is concerned, is it lower or higher than that of GLS1?
Answer: Obtaining an apparent Km value for aminotransferases that can be extrapolated back to the in vivo situation is tricky. Because aminotransferase (aka transaminases) catalyze a Ping-Pong type of reaction the apparent Km for an amino acid will increase as the concentration of the α-keto acid is increased. Nevertheless, approximate values for the Km for glutamine are in the 1 to 3 mM range for both rat GTK and GTL (Cooper and Meister Biochemistry 11, 661-671, 1972; J. Biol. Chem. 249, 2554-2561, 1974). In addition, it is now known that kynurenine aminotransferase 1 (KAT1) is identical to GKT and KAT3 is now known to be identical to GTL. Human KAT1/GTK has been reported to have an apparent Km of 2.8 mM toward glutamine (Han et al. Eur. J. Biochem. 271, 4804-14, 2004). Mouse KAT3/GTL has been reported to have a Km value toward glutamine of 0.7 mM (Han et al. Mol. Cell. Biol 29, 784-93, 2009). Inasmuch as glutamine is the most abundant amino acid in the body (Cruzat et al. Nutrients 10, E1864, 2918) and the concentration of glutamine in most tissues is mM (for example the concentration of glutamine in a normal 70 kg man has been estimated to be ~7 mmol/kg (~8 mM); Newsholme and Parry-Billings J. Parenter. Nutr. 14, 635-675, 1990) it would seem that, at first glance, the glutamine transaminases are working in vivo at quite high efficiency with glutamine. However, both enzymes have a broad amino acid specificity (Han et al. 2004, 2009) that may dilute the in vivo activity with glutamine. Nevertheless, the inhibition of turnover of glutamine by these amino acids may be relatively small, because of all the amino acids tested the highest catalytic efficiency with both enzymes is toward glutamine, and as noted above, the concertation of glutamine is higher than that of other amino acids. There is also one other consideration – both GTK and GTL are present in the cytosol and mitochondria whereas GLS1 and 2 are present mostly in the mitochondria. Botman et al (J. Histochem. Cytochem. 62, 813-826, 2014) reported a Km value for glutamine exhibited by glutaminase in mouse kidney and brain (i.e. for the GLS1 isoenzyme) of 0.6 mM and that for glutamine exhibited by glutaminase in mouse liver (i.e. for the GLS2 isoenzyme) of 11.6 mM.
The above discussion indicates that the affinity of the glutamine transaminases toward glutamine is comparable to that exhibited by the two glutaminase isoenzymes, but because of the factors outlined above it is not easy to draw conclusions about the relative quantitative importance of the glutaminase I and II pathways to glutamine metabolism solely from kinetic data obtained with purified enzymes or with tissue homogenates. However, we believe that tracer studies are relevant to this issue. Thus Darmaun et al (Am. J. Physiol. 251, E117-E126, 1986) administered [15N]glutamate, [2-15N]glutamine, [5-15N]glutamine and [2,5-15N]glutamine intravenously to adult human volunteers and carried out A-V kinetics of tracer disposition. From their findings the authors suggested that glutamine transamination may be more prevalent than is the glutaminase (PAG) reaction. These findings have been largely overlooked in the literature but are consistent with a prominent role for the glutaminase II pathway for the metabolism of glutamine in humans.
4) In the discussion, authors discuss the translational potential of glutaminase II pathway inhibitors? Do these inhibitors exist? Are they specific for these enzymes or also interact with other metabolic pathways involved in Gln metabolism?
Answer: Yes, inhibitors of the glutamine transaminases are known. For example, Ogier et al. (Biochem. Pharmacol. 25, 1631-44, 1993) and Quash et al. (Bull. Cancer 91, E61-79, 2004) showed that analogues of substrates bound to cofactor inhibit KMB metabolism in cells in culture. Inasmuch as KMB is a good substrate of both GTL and GTK it is probable that these substrate-cofactor mimics inhibit the glutamine transaminases. In the Quash et al. study the inhibitor induced DNA strand breaks in HeLa cells typical of apoptotic cell death.
The pharmaceutical industry has been interested in the kynurenine aminotransferases as in vivo inhibitors of kynurenate formation (Nematollahi et al. Int. J. Mol. Sci. 17, E946, 2016) even though the rate of turnover of kynurenine to kynurenate catalyzed by the glutamine transaminases (GTK/KAT I and GTL/KAT 3) must be miniscule compared to that exhibited with glutamine (Cooper et al. Amino Acids 48, 1-20, 2016). Some inhibitors have been designed to inhibit GTK/KAT1 (based on kynurenine as substrate) with Ki values ranging from ~20 μM to ~1 mM (Nematollah et al. Int. J. Mol. Sci. 17, E946, 2016) but to our knowledge these have not been evaluated as ant-cancer agents or as glutamine antagonists. [The pharmaceutical industry has yet to focus on designing inhibitors of GTL/KAT 3]. Another possibility would be to inhibit ω-amidase. However, no potent and selective inhibitors of this enzyme have been described. We envisage that publication of our article will spur the development of such ω-amidase inhibitors.
Thus, curently, there is no specific ω-amidase inhibitor known unlike inhibitors of the classical glutaminase (GLS1) pathway such as CB-839 and BPTES. As for the possibility that such GLS1 inhibitors could also interact with other pathways such as the glutaminase II pathway, there is no specific information available in this regard.
Reviewer 2 Report
In the manuscript “High Levels of Glutaminase II Pathway Enzymes in Normal and Cancerous Prostate and their Proposed Role in Glutamine Addiction”, Dorai et al. present data showing that GLS1 and glutaminase II pathway enzymes are highly expressed in normal prostate tissue and their expression increases with prostate cancer cells aggressiveness. Despite the data presented are clear, they are very limited and correlative. No data concerning possible roles and function of these enzymes in prostate cancer progression are presented. Moreover, there is an absolute disproportion between the very limited set of data presented, the discussion and the conclusions drawn by the authors. The title itself is misleading, since no results about the relationship between glutamine addiction in prostate cancer cells and the enzymes under investigation are shown.
Author Response
Reviewer 2
In the manuscript “High Levels of Glutaminase II Pathway Enzymes in Normal and Cancerous Prostate and their Proposed Role in Glutamine Addiction”, Dorai et al. present data showing that GLS1 and glutaminase II pathway enzymes are highly expressed in normal prostate tissue and their expression increases with prostate cancer cells aggressiveness. Despite the data presented are clear, they are very limited and correlative. No data concerning possible roles and function of these enzymes in prostate cancer progression are presented. Moreover, there is an absolute disproportion between the very limited set of data presented, the discussion and the conclusions drawn by the authors. The title itself is misleading, since no results about the relationship between glutamine addiction in prostate cancer cells and the enzymes under investigation are shown.
Answer: We thank this reviewer for his/her objective comments. The reviewer makes the comment that “Despite the data presented are clear, they are very limited and correlative”. We are sorry that the reviewer got this impression. The findings of high glutaminase II enzyme activities in the rat prostate is a new and important contribution to understanding amino acid homeostasis in the prostate. Admittedly the increase in intensity of staining of GLS1, GTK and ω-amidase on western blots as the Gleason grade increases is descriptive and semi-quantitative, but we believe that this is an important finding nonetheless. We have rewritten the text to indicate that although the findings are correlative they are consistent with the increasing importance of glutamine addiction in promoting the aggressiveness of the tumors.
Regarding the Title, we have removed the word “Proposed” and included the word “suggest” to properly describe all these possibilities that could be potentially therapeutic.
Reviewer 3 Report
Dorai et al. submit a study entitled „High levels of glutaminase II pathway enzymes in normal and cancerous prostate and their proposed role in „glutamine addiction“. In their well-written manuscript the authors show convincingly that the expression of the glutaminase II pathway increases with increasing aggressiveness of prostate cancer cells. To this end they use an interesting model system consisting of the increasingly metastatic cell lines C4, C4-2 and C4-2B previously isolated from LNCaP cells by the group around L.W. Chung. Western blot analyses of GLS1, omega-amidase, and GTK were performed on lysates of the cell lines, immunohistochemistry was done on human prostate and prostate cancer tissues, the enzyme activities were determined in fractions from homogenized rat tissues. The experimental approaches are sound and appropriate, the experiments appear to have been carried out cautiously and thoroughly, and the results are clear. With the statement on page 13, ll. 452-454: “The increased expression of the glutaminase II pathway enzymes in the stromal cell compartment may be a mechanism contributing to activation of this compartment to produce cancer associated fibroblasts (CAFs)”, the authors underline the potential of their study. The paper represents a significant contribution to the knowledge of the derailed metabolism in (prostate) cancer cells and may provide the basis for the development of new therapeutic strategies. Therefore, it should be of interest to a wide range of readers including pharmacologists, biochemists, oncologists and physiologists.
I have only two points that should be taken care of prior to publication:
-Figs. 4a and 4b: The cycles including the structural formulas and their labeling are too small even though the authors’ reasonable attention might be to draw the reader’s attention to the differences (between a=normal and b=malignant).
-Page 12, line 415 “..almost all such studies of cancer addiction in cancer cells assume…”. Probably, there is a typo in that the first “cancer” should probably be “glutamine”.
Author Response
Reviewer 3
Dorai et al. submit a study entitled „High levels of glutaminase II pathway enzymes in normal and cancerous prostate and their proposed role in „glutamine addiction“. In their well-written manuscript the authors show convincingly that the expression of the glutaminase II pathway increases with increasing aggressiveness of prostate cancer cells. To this end they use an interesting model system consisting of the increasingly metastatic cell lines C4, C4-2 and C4-2B previously isolated from LNCaP cells by the group around L.W. Chung. Western blot analyses of GLS1, omega-amidase, and GTK were performed on lysates of the cell lines, immunohistochemistry was done on human prostate and prostate cancer tissues, the enzyme activities were determined in fractions from homogenized rat tissues. The experimental approaches are sound and appropriate, the experiments appear to have been carried out cautiously and thoroughly, and the results are clear. With the statement on page 13, ll. 452-454: “The increased expression of the glutaminase II pathway enzymes in the stromal cell compartment may be a mechanism contributing to activation of this compartment to produce cancer associated fibroblasts (CAFs)”, the authors underline the potential of their study. The paper represents a significant contribution to the knowledge of the derailed metabolism in (prostate) cancer cells and may provide the basis for the development of new therapeutic strategies. Therefore, it should be of interest to a wide range of readers including pharmacologists, biochemists, oncologists and physiologists.
Answer: We thank the reviewer for the encouraging comments
I have only two points that should be taken care of prior to publication:
-Figs. 4a and 4b: The cycles including the structural formulas and their labeling are too small even though the authors’ reasonable attention might be to draw the reader’s attention to the differences (between a=normal and b=malignant).
Answer: We have also noted the same problem with the depiction of the metabolic cycles that include several structural formulae. Because of the condensed nature of this Figure, they tend to become too small and hence difficult to read.
To correct this, we have expanded the Figures to a larger scale so that the readers can now view and appreciate the biochemical alterations that go on in normal versus cancerous prostate.
-Page 12, line 415 “..almost all such studies of cancer addiction in cancer cells assume…”. Probably, there is a typo in that the first “cancer” should probably be “glutamine”.
Answer: In Page 12 and in line 415, the typo has been corrected.
Round 2
Reviewer 2 Report
The manuscript has not been improved. The authors did not make any of the changes suggested and the discussion is even longer than in the previous version of the manuscript.